# Immunosurveillance and molecular detection of hepatitis B virus infection amongst vaccinated children in the West Gonja District in Savanna Region of Ghana

Theophilus Quaye[1], Patrick Williams Narkwa[1]*, Seth A. Domfeh[2,3‡], Gloria Kattah[1,4‡], Mohamed Mutocheluh[1]

1 Department of Clinical Microbiology, School of Medicine and Dentistry, Kwame Nkrumah University of Science and Technology, Kumasi, Ghana, 2 Department of Biochemistry, Cell and Molecular Biology, School of Biological Sciences University of Ghana, Legon, Accra, Ghana, 3 West African Centre for Cell Biology of Infectious Pathogens, University of Ghana, Legon, Accra, Ghana, 4 Department of Medical Laboratory Technology, Radford University College, Accra, Ghana

☯ These authors contributed equally to this work.
‡ These authors also contributed equally to this work
* pnarkwa.chs@knust.edu.gh, patricknarkwa1@gmail.com

**Data Availability Statement:** The data are available in the Kwame Nkrumah University of Science and

## Abstract

Hepatitis B vaccination is the most effective preventive measure in reducing the incidence of chronic hepatitis B virus (HBV) infection and its consequences such as cirrhosis, hepatocellular carcinoma, liver failure and death. Ghana introduced the universal HBV vaccination in the national Expanded Programme on Immunization in 2002. The current study sought to determine the sero-protection rate and the prevalence of HBV infection among fully vaccinated children in the West Gonja District in the Savanna Region of Ghana. This cross-sectional study recruited three hundred and fifty (350) fully vaccinated children who visited West Gonja Catholic Hospital from September to December 2019 for healthcare. Structured questionnaires were administered to obtain information on the demographics. The clinical history of the participants was obtained from the hospital records. Sera were separated from 2-5ml of blood sample collected from each participant after informed consent had been sought from their parents/guardians. Sera were tested for HBsAg, anti-HBs and anti-HBc using ELISA. Samples positive for HBsAg or anti-HBc were tested for HBV DNA by Real-Time Polymerase Chain Reaction. The overall sero-protection rate (anti-HBs titers $\geq$ 10 mIU/mL) among the studied participants was 56% with anti-HBs geometric mean titer (GMT) of 95.7 mIU/mL ($\pm$ 6.0; 95% CI) compared with GMT of 2.8 mIU/mL ($\pm$ 0.2; 95% CI) among non-seroprotected participants. There was no statistically significant difference in sero-protection rate between males and females (p-value = 0.93) and in relation to age (p-value = 0.20). The prevalence of HBV infection among studied participants as determined by the HBV DNA/HBsAg positivity was 1.4% while anti-HBc sero-positivity was 2%. Even though the sero-protection rate and HBV infection rate reported in the current study compares with that of other international studies further studies need to be conducted to understand the factors related to sero-protection and HBV infection rate in the Savanna Region of Ghana.

Technology (KNUST) space repository: http://ir.knust.edu.gh/handle/123456789/14537.

**Funding:** The authors received no specific funding for this work.

**Competing interests:** The authors declare that they have no competing interests.

## Introduction

Even though hepatitis B virus (HBV) infection is a vaccine-preventable disease, the burden of chronic HBV infection in the world is still high especially in the sub-Saharan African Region [1]. It is estimated that approximately 257 million people worldwide are chronically infected with HBV with about 88% of them residing in sub-Saharan Africa (SSA) [2]. Chronic HBV infection has been identified as one of the significant risk factors in developing cirrhosis and hepatocellular carcinoma (HCC) [3]. In Ghana, the prevalence of chronic HBV infection has been estimated to be greater than 8% [4]. The best and the most effective way of preventing the perinatal and horizontal spread of HBV infection is through active vaccination. In view of that, World Health Organization (WHO) in 1992 recommended that all countries across the globe incorporate HBV vaccination into their national childhood immunization programme and by 2014, 184 countries had adopted this measure [5]. Ghana introduced hepatitis B vaccination as part of her Expanded Programme on Immunization (EPI) in 2002 [6]. Upon completion of the full doses of the vaccination series, the vaccine is expected to induce protective antibody levels in more than 95% of the recipients [7]. However, unreliable power supply in certain parts of sub-Sahara Africa including Ghana may affect the storage condition of the vaccine and hence its potency and efficacy. Administration of ineffective or impotent vaccine can lead to vaccination failure. Vaccination failure can predispose vaccinees to HBV infection and increase the transmission rate within the population. In view of this, it is appropriate that post-vaccination surveillance is carried out to assess the effectiveness of HBV vaccination programme in Ghana. Most studies in relation to HBV infection in Ghana have been focused largely on the prevalence of HBV among the unvaccinated component of the population. Even the limited studies that have been done on the vaccinated group since the commencement of the EPI in Ghana in 2002 have largely been conducted in the southern and the middle belt of the country and that data from Savanna Region of Ghana where West Gonja District is located is lacking hence the current study. It is anticipated that the outcome of the current study in the West Gonja District in the Savanna Region of Ghana would provide data or information that would help in assessing the effectiveness of hepatitis B vaccination programme in the District. This data would help policy makers in instituting measures that would detect early vaccination failures so that appropriate action could be taken to prevent spread of HBV infection in the district and Ghana as a whole.

## Materials and methods

### Study design and study site

This was a cross-sectional study conducted at the West Gonja Catholic Hospital, Damongo in the Savanna Region of Ghana from September to December 2019.

Damongo is the capital of the West Gonja District in the Savanna Region of Northern Ghana. The West Gonja District is located to the west of the Northern Regional capital Tamale. It lies within longitude $1^0 5^1$ and $2^0 58^1$ West and latitude $8^0 32^1$ and $10^0 2^1$ North. The West Gonja District shares boundaries to the south with Central Gonja District, Bole and Sawla-Tuna-Kalba Districts to the west, Wa East District to the north-west and North Gonja District to the east. The District has a total land area of 4715.9sqkm, part of which is occupied by the Mole National Park and Kenikeni Forest Reserves [8]. According to the 2010 Population and Housing Census, the population of West Gonja District is estimated at 41,180 which constitutes 1.7% of the total population in the Savanna Region. Generally, temperatures in the West Gonja District are high. The maximum temperatures are mostly recorded during the dry season between March and April while lowest temperatures are recorded between December and

January. The mean monthly temperature is 27˚C. The dry season is characterized by the Harmattan winds which are dry, dusty and cold in the morning and very hot at noon. During the Harmattan periods, the humidity of the West Gonja District becomes very low and this results in dry skin and cracked lips in humans. The area experiences bimodal rainfall pattern with annual rainfall being estimated at 1,144 mm. The rains begin in April and ends in late October. The peak of rainfall is in June and July with prolonged period in August [8].

## Study population and eligibility criteria

The study population comprised children aged 9 months to 17 years who had received the three doses of the pentavalent hepatitis B vaccine (DPT+HepB+ Hib) at ages 6, 10 and 14 weeks after birth as part of the EPI programme who visited the West Gonja Catholic Hospital, Damongo with their parents or legal guardians for healthcare. Children who had a complete record of vaccination duly endorsed in the vaccination cards were recruited into the study. The exclusion criteria included the following: (a) children who did not have proper vaccination records endorsed in their vaccination cards (b) children with interval between the last dose of hepatitis B vaccine and sampling less than one month and (c) children suffering from acute illness at the time of sampling. Using a prevalence (0.7%) of HBV infection among vaccinated Ghanaian children aged 5–32 months [9], a sample size of 350 was estimated at 95% confidence interval. Before their enrollment, written informed consent was obtained from the parents or legal guardians of the participants. Additionally, verbal assent was obtained from participants who were above 10 years. HBV vaccination of the participants was confirmed by taking a full detailed vaccination history from parents or legal guardians as well as examining the infant vaccination cards available with their parents. Fully vaccinated participants who declined or whose parents/legal guardians declined to partake in the study were excluded.

## Ethical consideration

The study protocol was reviewed and approved by the Committee on Human Research, Publications and Ethics (CHRPE) of School of Medicine and Dentistry (SMD), Kwame Nkrumah University of Science and Technology (KNUST), Kumasi-Ghana (reference number CHRPE/AP/555/19). Permission was sought from the management of the West Gonja Catholic Hospital, Damongo. Informed consent was sought from each parent or legal guardian of each participant after the purpose of the study had been explained to them in a language that they understood.

## Data and blood sample collection

A total of three hundred and fifty (350) participants were recruited into the study. Structured questionnaires were administered to obtain demographic information of the participants. The clinical history of the participants was obtained from hospital records. Two to five milliliters (2–5 ml) of whole blood samples were collected from each participant by veni-puncture into pre-labelled BD Vacutainer with SST II Advance serum-separator gel (BD, United Kingdom). The samples were allowed to clot for 10–15 minutes and then centrifuged at 3500 rpm for 5 minutes to separate the sera. The sera were aliquoted into two (2) freshly labelled Eppendorf tubes and temporarily stored at -20˚C at the laboratory of the West Gonja Catholic Hospital before being transported to the Virus Research Laboratory at the Department of Clinical Microbiology, School of Medicine and Dentistry at Kwame Nkrumah University of Science and Technology where the samples were stored frozen at -20˚C until testing.

## Laboratory analysis

Serum samples were tested for HBsAg, anti-HBs and anti-HBc. Qualitative testing of HBsAg, anti-HBc and quantitative testing of anti-HBs was performed using enzyme-linked immuno-sorbent assay (ELISA) (Fortress Diagnostics, UK) following the instructions of the manufacturer. For the ELISA test, the samples were initially tested in singles. Samples that were initially reactive were retested in duplicates using the same ELISA test kits. Repeatedly reactive samples were considered positive for HBsAg, anti-HBs and anti-HBc. The sensitivity and specificity of the test kits were greater than 99.0% as indicated by the manufacturer. Serum samples positive for HBsAg or anti-HBc were quantitatively tested for HBV DNA by Real-Time Polymerase Chain Reaction (RT-PCR) using a fully automated system at Komfo Anokye Teaching Hospital (KATH) in Kumasi in the Ashanti Region of Ghana. A fully automated HBV DNA extraction and RT-PCR amplification were done using COBAS® AmpliPrep Instrument (Roche Diagnostics, USA) and Cobas AmpliPrep/Cobas TaqMan (CAP/CTM) HBV test kits, v2.0 following the instructions of the manufacturer. A known HBV DNA positive sample was used as positive control while nuclease free water was used as negative control. The positive and negative controls were included in each run. The diagnostic sensitivity of the test kit was $\geq 95\%$ while the diagnostic specificity was 100% with a confidence limit of 99.54% as indicated by the manufacturer of the test kit. The thermal cycling conditions were set according to the guidelines of the manufacturer.

## Data analysis

Data generated were analyzed using GraphPad Prism version 8 software and IBM Statistical Package for Social Sciences version 20 software (IBM SPSS version 20). Because of the skewed distribution of data, anti-HBs geometric mean titer (GMT) was calculated to estimate the centrality of the anti-HBs level. Anti-HBs titer value of 0.05 mIU/mL was assigned to participants who had undetectable anti-HBs titer [10]. Categorical variables were expressed as frequencies and percentages. Unpaired students t-test was used to compare two (2) means while one-way analysis of variance (ANOVA) was used to compare more than two means. $p$-value $\leq 0.05$ was considered statistically significant. Participants with anti-HBs titers $\geq 1$ mIU/mL were considered to have sero-converted and vice visa as per kit manufacturer's instructions and WHO standard. Participants with anti-HBs titers $\geq 10$ mIU/mL were considered to be sero-protected against HBV infection.

## Results

### Sero-protection status of study participants in relation to demographic and clinical history

A total of 490 children who visited the West Gonja Catholic Hospital with their parents or legal guardians for healthcare during the study period were screened for eligibility out of which 26.5% (130/490) did not meet the inclusion requirements and were excluded from the study while 73.5% (360/490) met the inclusion requirements. Of the 130 children who did not meet the inclusion requirements, 57.7% (75/130) had never been vaccinated against HBV infection while 42.3% (55/130) had missing vaccination records. Of the 360 children who met the inclusion criteria, 27.8% (10/360) declined to partake in the study and were excluded. In all 350 children comprising 49.1% (172/350) males and 50.9% (178/350) females were recruited into the study. The mean age of the participants was 6.8 years (± 4.32 at 95% CI). Of the 350 participants tested for anti-HBs titer levels, sero-protection (anti-HBs titer $\geq 10$ mIU/mL) was detected among 56% (196/350) with geometric mean titer (GMT) of 95.7 mIU/mL (± 6.0; 95%

CI) compared to mean anti-HBs titer of 2.8 mIU/mL (± 0.2; 95% CI) among non-seroprotected participants (anti-HBs titer < 10mIU/mL). Of the 196 participants who were sero-protected, 52.0% (102/196) were males with GMT of 94.6 mIU/mL (± 8.2; 95% CI) while 48.0% (94/196) were females with GMT of 95.6 (± 8.7; 95% CI). There was no statistically significant difference in the sero-protection rate between males and females (p-value = 0.93). Majority (55.1%; 108/196) of the sero-protected participants were aged ≤ 5years with a GMT of 100.2 mIU/mL (± 7.6; 95% CI). However, there was no statistically significant difference in sero-protection levels among the different age groups (p-value = 0.20) Table 1.

### Sero-protection status of study participants in relation to clinical history

There was no statistically significant difference in sero-protection rate between participants with a history of hospital admission, open abscess, surgical operation, blood transfusion and participants with no such history (p-value = 0.29) Table 2.

### Molecular detection of HBV infection among the fully vaccinated participants

The prevalence of HBV infection as determined by HBsAg/HBV DNA positivity was 1.4% (5/350) while anti-HBc positivity was 2.0% (7/350). Of the 5 participants who tested positive for HBV DNA and HBsAg, 60.0% (3/5) were males while 40.0% (2/5) were females. In addition, 60.0% (3/5) of the participants who tested positive for HBV DNA and HBsAg were within the age group of 11–15 years. The majority of the participants who tested positive for anti-HBc were males (57.1%; 4/7) and within the age group of 11–15 years (57.1%; 4/7). Of the 7 participants who tested positive for anti-HBc, 71.4% (5/7) were positive for HBsAg and HBV DNA. The two (2) participants who were anti-HBc positive but HBsAg negative [anti-HBc (+ve)/HBsAg (-ve)] did not have detectable HBV DNA. All the five (5) participants who tested positive for HBsAg and the seven (7) participants who tested positive for anti-HBc had anti-HBs titer of < 10mIU/mL Table 3.

## Discussion

Reduction in transmission and the global burden of HBV is an achievable public health goal owing to the availability of commercially effective and safe hepatitis B vaccines. WHO in 1992 recommended that all countries incorporate the universal hepatitis B vaccination into their national childhood immunization programmes and by 2014, 184 countries had adopted this

**Table 1. Association of demographics and sero-protection level.**

| Variables | Anti-HBs (mIU/mL) | | | | | |
| --- | --- | --- | --- | --- | --- | --- |
| | < 10 mIU/mL | | | ≥ 10 mIU/mL | | |
| | n (%) | GMT (mIU/mL) (95% CI) | p-value | n (%) | GMT (mIU/mL) (95% CI) | p-value |
| Age (yrs.) | | | | | | |
| ≤ 5 | 39 (25.3) | 3.5 (± 0.46) | 0.11 | 108 (55.1) | 100.2 (± 7.6) | 0.20 |
| 6–10 | 52 (33.8) | 2.9 (± 0.42) | | 60 (30.6) | 99.5 (± 12.4) | |
| 11–15 | 60 (39.0 | 2.1 (± 0.34) | | 25 (12.8) | 61.8 (± 11.8) | |
| > 15 | 3 (1.9) | 5.0 (± 2.65) | | 3 (1.5) | 135.7 (± 69.1) | |
| Gender | | | | | | |
| Males | 70 (45.5) | 2.8 (± 0.34) | 1.00 | 102 (52.0) | 94.6 (± 8.2) | 0.93 |
| Females | 84 (54.5) | 2.8 (± 0.32) | | 94 (48.0) | 95.6 (± 8.7) | |
| Total | 154 (44.0) | 2.8 (± 0.20) | | 196 (56.0) | 95.7 (± 6.0) | |

**Table 2. Association of participants' clinical history and sero-protection level n (%) (p-value = 0.29).**

| Risk factors | n (%) | Levels of anti-HBs | | p-value | Odds ratio (95% CI) |
|---|---|---|---|---|---|
| | | < 10 mIU/mL n (%) | ≥ 10 mIU/mL n (%) | | |
| Hospital admission | | | | | |
| Yes | 219 (62.8) | 100 (45.7) | 119 (54.3) | 0.50 | 1.2 (0.8–1.8) |
| No | 131 (37.2) | 55 (42.0) | 76 (58.0) | | |
| Open abscess | | | | | |
| Yes | 16 (4.6) | 2 (12.5) | 14 (87.5) | 0.12 | 0.2 (0.0–0.8) |
| No | 334 (95.4) | 153 (44.1) | 181 (54.2) | | |
| Surgical operation | | | | | |
| Yes | 3 (0.9) | 2 (66.7) | 1 (33.3) | 0.44 | 2.5 (0.2–28.2) |
| No | 347 (99.1) | 153 (44.1) | 194 (55.9) | | |
| Blood transfusion | | | | | |
| Yes | 8 (2.3) | 1 (12.5) | 7 (87.5) | 0.10 | 0.2 (0.0–1.4) |
| No | 342 (97.7) | 154 (45.0) | 188 (55.0) | | |

measure [5]. Ghana introduced hepatitis B vaccination as part of her Expanded Programme on Immunization (EPI) in 2002 [6]. The vaccine is administered to new borns who receive three doses of hepatitis B vaccine at age 6, 10 and 14 weeks after birth and all the participants recruited into the study received the same dose of the pentavalent vaccine with the same paediatric vaccination schedule at infancy. After completing the primary vaccination series, the immune system of the recipients induces protective antibody levels in more than 95% of the recipients and the protection may last for at least 20 years and possibly for life [7]. The current study sought to determine the prevalence of HBV infection among fully vaccinated children in the West Gonja District in the Savana Region of Ghana. The overall sero-protection rate among the studied participants was 56%. This is comparable to the sero-protection rate of 57.2% [11], 54% [12] but higher than the 39.7% [13] reported in Egyptian studies conducted

**Table 3. Distribution of serological and molecular markers of HBV in relation to demographics.**

| Variables | Frequency n (%) | HBV DNA n (%) | HBsAg n (%) | Anti-HBc n (%) |
|---|---|---|---|---|
| Age (yrs.) | | | | |
| ≤ 5 | 147 (42.0) | | | |
| 6–10 | 113 (32.3) | 2 (40.0) | 2 (40.0) | 3 (42.9) |
| 11–15 | 88 (25.1) | 3 (60.0) | 3 (60.0) | 4 (57.1) |
| > 15 | 2 (0.6) | | | |
| Gender | | | | |
| Males | 172 (49.1) | 3 (60.0) | 3 (60.0) | 4 (57.1) |
| Females | 178 (50.9) | 2 (40.0) | 2 (40.0) | 3 (42.9) |
| HBV DNA | | | | |
| Positive | 5 (1.4) | | | |
| Negative | 345 (98.6) | | | |
| HBsAg | | | | |
| Positive | 5 (1.4) | | | |
| Negative | 345 (98.6) | | | |
| Anti-HBc | | | | |
| Positive | 7 (2.0) | | | |
| Negative | 343 (98.0) | | | |

among vaccinated children. Also, the 56% sero-protection rate reported in the current study compares favorably with the 56.7% reported among Iranian children [14] but lower than the 86.8% reported in South Africa [15], 90.0% reported in Brazil [16] and 88.7% reported in Bangladesh [17]. It was observed in the current study that the anti-HBs levels declined with increasing age. This finding is consistent with the findings of other studies [18,19]. Even though anti-HBs levels declined with increasing age, the difference observed in the current study was not statistically significant (p-value = 0.20). Additionally, no statistically significant difference in sero-protection rate was observed between males and females (p-value = 0.93). Some researchers have the opinion that a decline in the anti-HBs level below 10mIU/mL could make the individuals vulnerable to HBV infection and that a booster dose of HB vaccine may be needed [20]. However, other researchers are of the view that irrespective of the gradual fall and loss in the anti-HBs level, if the primary hepatitis B vaccination is adequately performed in healthy individuals, long term protection against HBV infection could be guaranteed and that booster dose may not be needed [21,22]. A study conducted in Egypt reported age and gender as the two main risk factors associated with non-sero-protection and that the risk was significantly higher in girls than boys [11]. It was observed in the current study that there was no statistically significant difference in non-sero-protection levels among the different age groups (p-value = 0.11) and between males and females (p-value = 1.0), a finding which is consistent with that of studies conducted in the USA [23] and Iran [24] but at variance with the findings of Salama et al.[11].

In the current study, it was observed that there was no statistically significant difference in sero-protection rate between participants with a history of hospital admission, open abscess, surgical operation, blood transfusion and participants with no such history (p-value = 0.29) a finding which is at variance with that of Salama et al. [11] who reported that children with the above-mentioned clinical history had significantly higher non-sero-protective rates compared to children with no such history (p-value<0.001). HBV infection was not detected among participants ≤ 5 years suggesting that there was no perinatal infection. This observation compares favorably with the findings of Soliman et al. [25] who reported that none of the Egyptian children aged < 5 years tested positive for HBsAg but another Egyptian study reported a prevalence of 0.8% among children aged ≤ 6 years [26]. However, 1.4% of the participants aged between 6–15 years were positive for HBsAg/HBV DNA while 2% of the children within the same age bracket were anti-HBc positive. All the participants who were HBsAg/HBV DNA positive were anti-HBc positive. The two participants who were anti-HBc positive but HBsAg negative [anti-HBc (+ve)/HBsAg (-ve)] did not have detectable HBV DNA indicating that there was no presence of occult HBV infection. Three mothers of the five participants who tested positive for both HBsAg and anti-HBc [anti-HBc (+ve)/HBsAg (+ve)] did not know their HBV status (data not shown) and therefore we could not tell whether the infected children contracted the HBV infection perinatally or horizontally. However, two mothers of the five participants who tested positive for both HBsAg and anti-HBc [anti-HBc (+ve)/HBsAg (+ve)] were negative for HBsAg/HBV DNA (data not shown) indicating that the participants might have contracted the HBV infection horizontally. All the five participants who tested positive for HBsAg and the seven participants who tested positive for anti-HBc had anti-HBs titer of < 10mIU/mL indicating that they were not sero-protected. The 1.4% HBsAg sero-positivity rate reported in the current study compares favorably with the findings of Wu et al. [27] who reported an HBsAg sero-positivity rate of 1.5% among vaccinated children in China. HBsAg sero-positivity rate reported in the current study was higher in males than females, although not statistically significant. The finding is consistent with that of Odusanya et al. [28] who reported that there was no statistically significant relationship between HBsAg sero-

positivity rate and gender. The 2.0% anti-HBc sero-positivity rate reported in the current study compares favorably with the findings of other studies [29].

## Conclusion

From this study, it was observed that 56% the study participants attained sero-protection status after primary hepatitis B vaccination but 44% of the participants who had been fully vaccinated against HBV still remain unprotected and could be susceptible to HBV infection. The prevalence of HBV infection as determined by HBsAg/HBV DNA positivity was 1.4% while anti-HBc sero-positivity was 2%. Further studies need to be conducted to understand the factors related to the sero-conversion and sero-protection rate in the West Gonja Municipal in the Savanna Region of Ghana. Additionally, further studies need to be conducted across the country to assess the effectiveness of HB vaccination in the country.

## Limitations of the study

This study did not evaluate the effectiveness or efficacy of the hepatitis B vaccine but rather evaluated the ability of hepatitis B vaccination in providing protection thereby reducing the susceptibility of the vaccinated participants to the HBV. Also, age-matched unvaccinated controls residing in the same area were not recruited into the study to enable us determine whether the prevalence of HBV infection was the same in vaccinated and unvaccinated group so as to determine the efficacy of the vaccine in the current study.

## Supporting information

**S1 File. Questionnaire used in the study.**
(PDF)

## Acknowledgments

We wish to sincerely thank all those who contributed to developing this new research, particularly, the research participants and those who helped with recruitment, sample collection, laboratory testing and data entry. Eddie-Williams Owiredu, Isaac Kusi-Amponsah, Hamdiyat Mustapha M-asheda, Alimatu Braimah and Alhassan Amina.

## Author Contributions

**Conceptualization:** Theophilus Quaye, Patrick Williams Narkwa, Mohamed Mutocheluh.

**Formal analysis:** Patrick Williams Narkwa, Mohamed Mutocheluh.

**Funding acquisition:** Patrick Williams Narkwa, Mohamed Mutocheluh.

**Investigation:** Theophilus Quaye, Seth A. Domfeh, Gloria Kattah.

**Project administration:** Seth A. Domfeh, Gloria Kattah.

**Resources:** Theophilus Quaye.

**Supervision:** Patrick Williams Narkwa, Mohamed Mutocheluh.

**Writing – original draft:** Theophilus Quaye.

**Writing – review & editing:** Patrick Williams Narkwa, Seth A. Domfeh, Gloria Kattah, Mohamed Mutocheluh.

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
