## [Decision Letter · Decision Letter 0]

5 Jul 2021

PONE-D-21-12148

Immunosurveillance and molecular detection of hepatitis B virus infection amongst vaccinated children in the West Gonja District in Savanna Region of Ghana

PLOS ONE

Dear Dr. Narkwa,

Thank you for submitting your manuscript to PLOS ONE. After careful consideration, we feel that it has merit but does not fully meet PLOS ONE’s publication criteria as it currently stands. Therefore, we invite you to submit a revised version of the manuscript that addresses the different points raised during the review process.

We look forward to receiving your revised manuscript.

Kind regards,

Isabelle Chemin, PhD

Academic Editor

PLOS ONE

Journal Requirements:

Reviewers' comments:

Reviewer's Responses to Questions

**Comments to the Author**

1. Is the manuscript technically sound, and do the data support the conclusions?

Reviewer #1: Yes

2. Has the statistical analysis been performed appropriately and rigorously? 

Reviewer #1: Yes

3. Have the authors made all data underlying the findings in their manuscript fully available?

Reviewer #1: Yes

4. Is the manuscript presented in an intelligible fashion and written in standard English?

Reviewer #1: Yes

5. Review Comments to the Author

Reviewer #1: This is a cross-sectional study conducted by investigators at three universities of Ghana to decide the seroprotection rate and the prevalence of hepatitis B virus (HBV) infection among 350 fully vaccinated children in the West of Gonja, Ghana, Africa, in 2019. Ghana is a low-income country where HBV vaccination was implemented in 2002. The study supplies data like international publications.

Although hepatitis B is an immuno-preventable disease, there is still a substantial proportion of people suffering from chronic infection and of these, more than 80% are in sub-Saharan Africa. Although in Ghana the prevalence of chronic HBV infection is less than 10%, it is particularly important to know the vaccination status.

The authors presented an interesting manuscript, whose implemented method that supports the obtained findings and their conclusions. The results of this study show the importance of monitoring seroprotection after vaccination for HBV to assess the ability of the vaccine.

This study is also important because in West Gonja, Damongo no studies have been carried out to evaluate seroprotection and therefore its findings supply new knowledge. Although electricity problems are known to affect the quality of the vaccine, it is important that these findings are considered when implementing post-vaccination surveillance programs.

Specific comments

1. The introduction has the overall context and approach of importance of vaccination HBV.

2. In the study design and study site section, the authors should include a further description of the geographical area where the study was carried out. Consider aspects of temperature, organization, size of the city, population or another information that authors considers important.

3. In the study population and eligibility criteria to include how much is the n of the population from 9 months to 17 years and of these how many are fully vaccinated. What was the sample size calculation? Indicate the prevalence for hepatitis B infection and the power considered to calculate the sample size. Additionally, describe in detail the inclusion criteria for the study.

4. In the laboratory analysis section to include the number of replicates of the experiment and how the authors confirmed the results.

5. In the DNA detection section, which controls were used. Mention the sensitivity and specificity of the molecular test.

6. In the results section, explain in detail how many children were initially screened and how many didn't have the inclusion criteria.

7.It is suggested to the authors for the presentation of the results of Table 3, to make a flow chart that writes down in better detail the results obtained in the serological and molecular study. This table does not have an adequate presentation.

8. The conclusion describes that 86% of the participants seroconverted after primary vaccination. Could the authors explain how was obtained this result?

6. PLOS authors have the option to publish the peer review history of their article (what does this mean?). If published, this will include your full peer review and any attached files.

Reviewer #1: No

---

## [Author Response · Author response to Decision Letter 0]

28 Jul 2021

20th July, 2021 

The Editor in Chief

PLoS One Journal

Dear Editor in Chief,

SUBMISSION OF REVISED MANUSCRIPT FOR PUBLICATION

I would like to submit a revised version of our manuscript titled ‘Immunosurveillance and molecular detection of hepatitis B virus infection amongst vaccinated children in the West Gonja District in Savanna Region of Ghana by Theophilus Quaye, Patrick W. Narkwa, Seth A. Domfeh, Gloria Kattah and Mohamed Mutocheluh to be considered for publication as a research article in the PLoS One Journal.

We thank the academic editor and the reviewer(s) for their generous comments on the manuscript. We have edited the manuscript to address their concerns. 

Below are the full responses to the comments of the academic editor and the reviewer(s).

Academic editor’s comments and responses

Comment 1

Response

The manuscript has been carefully checked against PLOS ONE’s style requirement and we believe these requirements have been duly met.

Comment 2

Please include additional information regarding the survey or questionnaire used in the study and ensure that you have provided sufficient details that others could replicate the analyses. For instance, if you developed a questionnaire as part of this study and it is not under a copyright more restrictive than CC-BY, please include a copy, in both the original language and English, as Supporting Information.

Response

The questionnaire that was developed and used in the study has been provided as supporting information.

Comment 3

 We note that you have included the phrase “data not shown” in your manuscript. Unfortunately, this does not meet our data sharing requirements. PLOS does not permit references to inaccessible data. We require that authors provide all relevant data within the paper, Supporting Information files, or in an acceptable, public repository. Please add a citation to support this phrase or upload the data that corresponds with these findings to a stable repository (such as Figshare or Dryad) and provide and URLs, DOIs, or accession numbers that may be used to access these data. Or, if the data are not a core part of the research being presented in your study, we ask that you remove the phrase that refers to these data.

Response

The sentence which was referenced as ‘data not shown’ has been deleted as it does not form a core part of the study.

Review comments and responses

Comment 1

The introduction has the overall context and approach of importance of vaccination HBV.

Response

Not Applicable

Comment 2

In the study design and study site section, the authors should include a further description of the geographical area where the study was carried out. Consider aspects of temperature, organization, size of the city, population or another information that authors considers important.

Response:

The study design and site section have accordingly been updated with the description as suggested by the reviewer in the revised manuscript.

Comment 3

In the study population and eligibility criteria to include how much is the n of the population from 9 months to 17 years and of these how many are fully vaccinated. What was the sample size calculation? Indicate the prevalence for hepatitis B infection and the power considered to calculate the sample size. Additionally, describe in detail the inclusion criteria for the study.

Response

Data on how much is the n of the population aged 9 months to 17 years was very difficult to obtain as data from Ghana Statistical Service 2010 Population and Housing Census only indicated that the population of the West Gonja District is youthful with14.6% being between the ages of 0-4 years. The prevalence of hepatitis B infection and the power considered in calculating the sample size has been indicated in the revised manuscript. Also the inclusion and exclusion criteria have been described.

Comment 4

In the laboratory analysis section to include the number of replicates of the experiment and how the authors confirmed the results.

Response

The number of replicates and how the ELISA tests results were confirmed have been indicated in the revised manuscript.

Comment 5

In the DNA detection section, which controls were used. Mention the sensitivity and specificity of the molecular test.

Response

The controls used in RT-PCR have been indicated in the revised manuscript. The sensitivity and specificity of the molecular test as indicated by the manufacturer of the kits have been stated.

Comment 6

In the results section, explain in detail how many children were initially screened and how many didn't have the inclusion criteria.

Response

The number of children who were screened and met the inclusion criteria have explained in the revised version of the manuscript.

Comment 7

It is suggested to the authors for the presentation of the results of Table 3, to make a flow chart that writes down in better detail the results obtained in the serological and molecular study. This table does not have an adequate presentation.

Response

The focus of Table 3 is to look at the distribution of serological and molecular markers of HBV in relation to the demographics of the study participants. Several options including graphical presentation as well as use of flow chart (as suggested by the reviewer) in presenting the data were considered initially. But after careful analysis, we observed that using graphs or flow chart in presenting the data would rather have some information hidden which would make it a bit difficult for readers to comprehend. We are therefore of the opinion that presenting the data in the table as has been done in Table 3 better makes it easier to understand the information that we want to put across.

Comment 8

The conclusion describes that 86% of the participants seroconverted after primary vaccination. Could the authors explain how was obtained this result?

Response 

The sero-conversion rate was determined by dividing the number of participants which had anti-HBs titers > 1mIU/mL by the total number of participants that is 301 divided by 350. However since the focus of the study was on the level of protection gained after hepatitis B vaccination, the sero-conversion part in the conclusion has been removed as the sero-conversion analysis was not captured under the result section. 

We believe the manuscript is now suitable for publication in PLoS One Journal.

Sincerely yours,

SIGNED

Patrick W. Narkwa (PhD)

(Corresponding author)

---

## [Editor Report · Decision Letter 1]

24 Aug 2021

Immunosurveillance and molecular detection of hepatitis B virus infection amongst vaccinated children in the West Gonja District in Savanna Region of Ghana

PONE-D-21-12148R1

Dear Dr. Narkwa,

We’re pleased to inform you that your manuscript has been judged scientifically suitable for publication and will be formally accepted for publication once it meets all outstanding technical requirements.

Kind regards,

Isabelle Chemin, PhD

Academic Editor

PLOS ONE
---

## [Editor Report · Acceptance letter]

8 Sep 2021

PONE-D-21-12148R1 

Immunosurveillance and molecular detection of hepatitis B virus infection amongst vaccinated children in the West Gonja District in Savanna Region of Ghana 

Dear Dr. Narkwa:

I'm pleased to inform you that your manuscript has been deemed suitable for publication in PLOS ONE. Congratulations! Your manuscript is now with our production department. 

Kind regards, 

on behalf of

Mrs Isabelle Chemin 

Academic Editor

PLOS ONE